# AMARO—An On-Board Ship Detection and Real-Time Information System

**DOI:** 10.3390/s20051324

**Published:** 2020-02-29

**Authors:** Katharina Willburger, Kurt Schwenk, Jörg Brauchle

**Affiliations:** 1German Space Operations Center (GSOC), German Aerospace Center (DLR), Münchener Straße 20, 82234 Weßling, Germany; kurt.schwenk@dlr.de; 2Institute of Optical Sensor Systems, German Aerospace Center (DLR), Rutherfordstr. 2, 12489 Berlin, Germany; joerg.brauchle@dlr.de

**Keywords:** real-time communication, maritime situational awareness, ship detection, Iridium, on-board, image processing, flight campaign

## Abstract

The monitoring of worldwide ship traffic is a field of high topicality. Activities like piracy, ocean dumping, and refugee transportation are in the news every day. The detection of ships in remotely sensed data from airplanes, drones, or spacecraft contributes to maritime situational awareness. However, the crucial factor is the up-to-dateness of the extracted information. With ground-based processing, the time between image acquisition and delivery of the extracted product data is in the range of several hours, mainly due to the time consumed by storing and transmission of the large image data. By processing and analyzing them on-board and transmitting the product data directly as ship position, heading, and velocity, the delay can be shortened to some minutes. Real-time connections via satellite telecommunication services allow small packets of information to be sent directly to the user without significant delay. The AMARO (Autonomous Real-Time Detection of Moving Maritime Objects) project at DLR is a feasibility study of an on-board ship detection system involving on-board processing and real-time communication. The operation of a prototype system was successfully demonstrated on an airborne platform in spring 2018. The on-ground user could be informed about detected vessels within minutes after sighting without a direct communication link. In this article, the scope, aim, and design of the AMARO system are described, and the results of the flight experiment are presented in detail.

## 1. Introduction

Nowadays, about 90% of the world’s volume of cargo is seaborne [1]. An enormous amount of money depends on reliable transportation routes. However, safeguarding the seaways is not only essential for the carriage of goods, but especially for the integrity of humans’ lives. Piracy, illegal fishery, ocean dumping, and refugee transportation are daily occurrences.

Due to these reasons, maritime surveillance is an important factor for government and private organizations. The European Maritime Safety Agency (EMSA), for example, has set up a vessel traffic monitoring and information system to be able to receive information on ships, ship movements, and hazardous cargoes [2]. General information around maritime domain awareness and how it is handled today can be found in [3,4].

One major issue of maritime surveillance is the vast expanse of the sea on the Earth’s surface, which makes observation of ship traffic difficult [5]. The only method to globally get general reliable information about a ship’s current position in near-real-time is by using satellite-based AIS (Automatic Identification System) information services [6]. AIS is a cooperative system, primarily intended for collision avoidance. Ships send out their identification, position, course, speed, and several other traffic-related data. This data is then received by other ships and ground stations in close range. Nowadays, to be able to track ships globally in real-time, satellites are also used to receive AIS data [7]. However, based on AIS data only, the detection of illegal activities like water pollution, illegal fishing, or smuggling is limited.

To improve maritime domain awareness, Earth observation (EO) satellite data is a valuable source of information. Great efforts are made in researching the potential of vessel detection in optical and radar satellite images [8,9,10]. However, in most cases, these images are analyzed long after the data have been acquired [11]. To tackle this bottleneck, there is also promising progress in establishing near-real-time services on the ground, which today can provide information in, at best, the range of 15 min, measured from on-ground data reception [12,13]. However, the most significant time delay occurs between data acquisition on-board and data reception on-ground, since image data are comparatively huge and their downlink requires a direct contact to a ground station. This delay can amount to hours or even days [14].

A second drawback of EO satellites for time-critical applications is their inability to continuously monitor a defined region of interest. Satellites with a reasonable spatial resolution for ship detection orbit in LEO (low Earth orbit) with speeds of approximately 7 km/s over ground, and typically have a revisit cycle of several days [15].

Promising upcoming observation platforms are unmanned autonomous vehicles [16]. For instance, with their Remotely Piloted Aircraft Systems (RPAS), the European Maritime Safety Agency operates a number of services supporting maritime surveillance [17]. These vehicles are small, lightweight, and ready to take off within minutes [16]. However, their operational flight duration, and thus the range of their geographical applicability, is limited. High-altitude pseudo-satellites (HAPS) are the perfect fit for long-endurance wide-area monitoring tasks. Although there is still a significant portion of development left, major progress has been achieved during recent years. One of the most famous HAPS, the Airbus Zephyr S, can carry a payload of up to 20 kg; with nearly 26 days, it holds the world record for the longest uninterrupted flight [18]. However, if HAPS shall be flexibly and rapidly deployable, even in remote areas, they have to overcome a similar problem to that of satellites: Downlinking time-critical information as fast as possible and informing the user immediately without a direct link to a ground station.

To reduce the time between acquiring data with an Earth observation (EO) platform and delivering meaningful information to the user, the capability of real-time communication from the satellite to the ground is needed. One option is using satellite communication services like Iridium or Orbcomm [19]. These services are able to transfer data 24/7 nearly globally within a few minutes, but offer only restricted bandwidth, which is insufficient to send the raw sensor data continuously down to a ground station for on-ground processing. However, the product data that should reach the user within the shortest possible time typically comprises small information like position, heading, velocity, type, and status of the ship. With on-board processing, this information can be extracted directly after acquisition. Since its size amounts to only a fraction compared to the raw sensor data, it can be sent to the user via the mentioned satellite communication services.

A challenge for on-board data processing is that of the limited computer resources that are available on satellites or other autonomous platforms. Furthermore, the special hardware that is used for on-board systems often differs from the mature technology in on-ground data centers, which makes it unfeasible to simply let the on-ground algorithm run on-board. This problem is discussed widely in the literature. In [20], Yuan Yao et al., present a computing system for on-board vessel detection targeting micro- and nano-satellites. This ship detection system extracts image patches and position information from acquisitions using deep learning methods, with the goal of decreasing data size. The authors were able to reduce an image with a size of 90 MB to product data below 1 MB within 1.25 s with a Commercial Off-the-Shelf (COTS) NVIDIA Jetson TX2. In [21], Yu Ji-yang et al., proposed a real-time on-board ship detection method based on FPGA hardware. They used statistical analysis and shape information for extracting images by marking their pixels. On an 8 bit image with 1024 × 1024 pixels, they were able to extract ships within 10 s with a precision and recall of over 90%.

Another question which seems to be disregarded so far is that of how a modern on-board computing information system should operate as a whole. With the on-board processing systems mentioned above, data are only analyzed and product data are sent to the ground. This approach is a static concept, not allowing user interaction. What we are targeting is an overall and more flexible system, where the user is able to order data, as is done in a web query. They should also be free to choose when and about what to be informed, and to be able to set automated alarms, which are pushed to them in the case of the occurrence of predefined events.

Within this paper, we present the results of a feasibility study of a comprehensive concept for a real-time on-board ship detection system for satellites and other kinds of unmanned flying vehicles. The study involves the development of a prototype system, called AMARO (Autonomous Real-Time Detection of Moving Maritime Objects), and its testing within an aircraft flight experiment campaign. The focus of the study was on how to design a flexible real-time ship detection system for on-board operation, how to realize it, and what performance, especially regarding real-time information capability, can be expected. The prototype system was designed and built using COTS hardware adequate for the aircraft test campaign. The prototype processes image data on-board and communicates the extracted information to the user immediately and without geographical bounds. The system provides product data like the position, heading, velocity, and shape of ships within minutes after sighting. Furthermore, this product data can be individually requested by the user via email on any smart device on the ground, independently of its locality. AMARO was tested in a flight experiment which took place in April 2018.

## 2. Materials and Methods

### 2.1. Conceptualization

The initial situation we assume involves an EO platform and a user on-ground who demands to be informed about ship-related events in the shortest possible time. Evaluating various usecase scenarios, the following requirements were identified:The user shall be able to post user-defined requests.The user shall be able to define events about which he/she is informed automatically.The user shall be able to get information with a topicality of at least five minutes.The communication to the user shall be location-independent (e.g., open sea).The information available shall include the object’s position, classification, shape attributes (e.g., size, perimeter), trajectory, and estimated heading and velocity.The information available shall include a small preview image of the object.

Requirements 1 and 2 demand a bidirectional communication link, where users can interactively exchange custom-tailored information with the on-board system. Requirements 3 and 4 imply that the images are processed on-board and messages are linked via a satellite-based communication system, since direct links are ineligible due to their limited range. Requirement 5 suggests the usage of a database for storing and managing information.

Based on these deliberations, the concept of the AMARO on-board ship detection system was developed. It consists of one or more Earth observing platforms carrying a camera, a GNSS receiver, an on-board computer, and a modem for real-time communication. An AIS receiver can be mounted on board, and its signals can be synchronized with the image data. Ships in the observation area, which send no signal—possibly on purpose—can thus be identified.

On board, ships are detected from the image data by means of remote sensing algorithms. Product data like position, heading, velocity, type, and status of the ship are extracted. These data—some kilobytes in size—can be sent from the EO platform to the network of communication satellites, which forwards the message until it can be delivered. A small quicklook of the detected object can be included for visual inspection. At most, this procedure will take a few minutes.

On operation, sensor data are acquired continuously by the camera system. These data are immediately evaluated on-board the flying platform, and the product data are stored in a database on the satellite. The user shall be able to query this database by using real-time communication. Furthermore, the user shall be able to define events about which he/she is automatically informed.

Figure 1 shows an exemplary sequence of events involving automatically transmitted and manually requested information: A user is interested in ships that send no AIS signals. He/she therefore requests to be informed automatically if a corresponding event occurs. He/she will not be spammed with information about other detected objects that do not fulfill his/her requirements. As soon as a ship without AIS is detected, a message is sent automatically from the Earth observing platform to the user via the satellite communication network. Since the user is further interested in this ship, he/she requests details via a one-time order. Among others, these details can include a small image of the detected object in order to verify it by visual inspection.

### 2.2. Hardware Architecture

The AMARO-Box and its contents were specially built for the airborne test campaign. The different hardware devices are therefore not necessarily suitable for an operation on another EO platform. An image of the box during assembly is shown in Figure 2. In the following, the components of the AMARO-Box are explained in detail. Since the camera, which was used in the experiment, and the corresponding image data are an essential part of the flight campaign, but not of the AMARO-Box, they are explained later on, in Section 2.4.

#### 2.2.1. Communication System

As mentioned before, the main goal of AMARO is generating and delivering the product information to the user as fast as possible. The user may be a crisis intervention center with a high-rate internet connection or a single person off any ground-based connection for communication. Similarly, the carrier platform of the AMARO-Box may be off any connection to ground-based communication facilities. Therefore, in order to facilitate permanent and locally independent communication, the use of satellite connections was considered mandatory. The following criteria for the real-time transmission of the product data have been collected: Low latency, global coverage, easy to obtain, easy to maintain, easy to integrate, and easy to operate.

After a careful deliberation over various options, we decided to use the Iridium Short Burst Data (SBD) service. Iridium is a satellite communication network consisting of 66 active satellites, which provides almost 100% global coverage continuously at almost 24/7. With SBD, Iridium offers a simple and efficient service for transmitting small data packages between equipment and centralized host computer systems [22], commonly used for asset tracking. Messages with a size of around 300 B can be exchanged between the on-board device and the on-ground user. For sending and receiving messages from the device, standard email is used. The email is sent to Iridium with the device’s serial number as the subject. The message itself is attached to the email as a normal text file with extension *.sbd and can be of individual content. In the case of AMARO, this *.sbd file contained a database query in the sql-language.

The latency for data exchange is specified as less than one minute worldwide [23]. The size of the transceiver device is ( 31.5 mm
× 29.6 mm
× 8.1 mm w/h/d). The average power consumption is below 0.8 W.

One big advantage of the Iridium system is that the antenna needs no exact pointing alignment to a determined direction. On an aircraft, it is sufficient that the antenna points approximately to the sky. This may not apply to other platforms. The satellites of the Iridium network fly in orbits of approximately 780 km height, and their signals are broadcast such that the regions of reception overlap on the Earth’s surface. However, a loss of coverage is supposable for platforms in higher altitudes.

The Iridium SBD modem and antenna are available custom off the shelf; hence, the purchase is fast and uncomplicated. We decided to buy a MiChroBurst-Q modem from Wireless Innovation [24]. It houses an Iridium 9602 modem and comes development-ready with connection ports for power supply and data transfer via RS-232. The whole box is sized 110 mm
× 35 mm
× 85 mm w/h/d, which is comparable to a packet of cigarettes.

As an Iridium antenna, we bought an AeroAntenna AT2775-110 [25]. Since the antenna had to be specially mounted on the plane’s roof, the owner required an aircraft-certified device and its installation by a specialist. The antenna is flat and streamlined to fulfill the aerodynamic requirements, as can be seen in Figure 3. It operates in the frequency band of (1595±30) MHz and consumes approximately 10 W.

While implementing and testing the AIS subsystem, we had access to an AIS simulator. With its ability to generate fake AIS messages that can be received by our system, validation was greatly eased.

The operation cost of the SBD service was a minor factor. In the time of operation, we paid around $20 for a monthly data volume of around 12 kB. This is depending on the size of the messages, equivalent to 40 to 120 messages.

#### 2.2.2. AIS Receiver

To able to receive AIS navigation data from accordingly equipped vessels, an AIS receiver and antenna were installed on the airplane. The receiver we used, AMTEC CYPHO-150, is a custom version of the shelf standard device, whose primary usecase is to be installed on recreational boats, which do not need to send out AIS information. The AMTEC CYPHO is not especially qualified for deployment on airplanes and is hence available at a fraction of the price of a dedicated device. However, it worked perfectly without any trouble in installation or loss in performance.

This AIS receiver is capable of receiving AIS messages of classes A and B [26], sent out by commercial and private vessels, respectively. Receiving several other AIS formats is also possible, but was not in our interests.

It is lightweight, small in form factor ( 128 mm
× 36 mm
× 88 mm w/h/d), and has a power consumption below 1.50 W; hence, it was perfectly suited to be installed in the AMARO computing box [27].

The AIS receiver can be connected to the on-board computer via a serial or USB interface. We chose the latter, because it can also be used to power the device. To encode AIS messages, the AMTEC CYPHO-150 uses a serial text-based transmitting protocol specified by the NMEA 0183 interface standard. Typically, AIS messages contain the Maritime Mobile Service Identity (MMSI) number, the call sing and name, the type, the length and beam, the cargo information, the position of the vessel, the Course Over Ground (COG), the Speed Over Ground (SOG), the heading, the speed of the ship, and the status of the ship. The AIS messages were parsed by our on-board software and then directly inserted in the database. Our parser was based on the libais library (see [28]) and modified to meet our needs.

As the AIS antenna, a standard PROCOM HX2 was used [29]. It is a flexible 1/4 λ helix antenna for the two AIS channels in the frequencies 161.975 and 162.025 MHz. It is around 150 mm long, and was installed together with the camera in the downward-looking hole of the aircraft’s fuselage. While implementing and testing the AIS subsystem, we had access to an AIS simulator. With its ability to generate fake AIS messages that could be received by our the system, validation was greatly eased.

#### 2.2.3. On-Board Computer

The on-board computer is the core component of AMARO. It obtains the camera data, controls the Iridium transceiver and the AIS receiver, performs data analysis, and manages inner and outer communication. The requirements for the on-board computer were the following: It had to be small to fit in a 19 inch rack box together with the other components. It had to provide sufficient computing power for data processing. Moreover, it had to be physically and thermally robust for reliable operation on the aircraft (passenger cabin).

After studying the market, we decided to buy a 1.3 l slim standard personal computer (Shuttle DQ170), which is equipped with standard up-to-date desktop PC components. The computer is robust enough to handle 24/7 operation and up to 50 ${}^{\circ }C$ ambient air temperature. All interfaces needed for attaching the other devices are present. Equipped with modern desktop PC components, that is, Intel Core i7-6700, 16 GB RAM, and a 512 GB SSD, the system may be luxurious compared to today’s or even future computer solutions, deployable on-board HAPS or satellites. Power usage, thermal output, space limitations, and radiation impact were insignificant for the demonstration of our prototype. Therefore, for the first proof of concept, we determined a restriction in this regard to be unnecessary. Nevertheless, since we are also involved in building a next-generation space-computing platform [30], we assume that it is possible to integrate the software on a future computer mounted on an autonomous carrier platform.

### 2.3. Software Architecture

The software is the most important and labor-intensive component of the AMARO system. Whereas most hardware components could be bought off the shelf, the software system has been developed from scratch. It is designed to be modular and flexible, such that it can modified for a variety of scenarios and deployed on an arbitrary carrier platform.

#### 2.3.1. Software Requirements

The AMARO software system has to handle two main tasks: Data analysis and communication. The data analysis process shall extract useful information from the image data or other sources. The system shall be capable of processing as much data as possible to reach a high situational awareness. Due to the complexity of the ship detection algorithm and the high amount of image data, processing may be computationally intensive. The AMARO communication system has to be readily responsive, and the available bandwidth has to be used efficiently. Since there are only limited maintenance options at runtime and the system is deployed on-board a flying platform, it has to be absolutely reliable. Interruptions of operations are undesired, and in the case of an error, the system has to mitigate it and get back to operation with the least possible loss of information.

#### 2.3.2. Software Infrastructure

To enable fast and efficient software development, a standard x86-64 Linux desktop distribution was selected as the operating system. As the main programming language, C++14 was chosen. Based on these conventions, a lot of up-to-date software development tools and libraries are available to help minimize development costs. The effort to deploy the software is further minimized, since the development and runtime operating systems are identical.

We want to stress that the goal of the development was to build a software system that proves the concept of a real-time on-board ship detection system within an experimental flight. Nevertheless, as C++ and Linux are also used for future on-board systems, we trust that our software is, in principle, implementable on an on-board platform without fundamental changes. In fact, we have already ported essential parts of our software to an on-board computer within the project ScOSA (Scalable On-Board Computing for Space Avionics), which has the goal of developing a high-performance on-board platform for the deployment on satellites [31].

#### 2.3.3. Software Design

As mentioned in Section 2.3.1, the system has to be high-performance, responsive, and reliable. To meet all of these requirements, a service-based architecture was chosen. A top-level view of the service architecture is presented in Figure 4. Every task is carried out by a unique service that can operate independently of other services. As every service is its own Linux process, high responsiveness for the communication services and, at the same time, a high amount of computation time for the ship detection application can be provided. In case of an error, aborting a service has no direct effect on other processes, and the service can be restarted individually.

For inter-service communication and for data storage, we chose the file-based database SQLite [32]. SQLite can be easily implemented without the need for a dedicated database server. The (asynchronous) communication of the services is handled by the database engine itself. Furthermore, the validity of the database is warranted by SQLite in case of a writing failure. The most functional advantage of using an SQL database is the availability of the language SQL (Structured Query Language). The SQL programming language is the key enabling element used for the implementation of the user interaction with the system. In general, SQLite is not recommend for distributed systems (e.g., network file systems) and is not well suited for heavy simultaneous writing on one database file. However, for the experimental demonstration of our prototype system, all services were located on the same computer, and data were written simultaneously to one database file only sparsely. For a future operational system, the usage of a server-based database is strongly recommended.

The following subsections describe the different independent services of the AMARO software.

#### 2.3.4. Service SBD Message

Our serviceSBD is a messaging service that allows other services to send and receive messages over the Iridum SBD. A service that wants to send a message adds it to the so-called *toSendMsg* table of the database. When a sending slot is available, serviceSBD checks this table and tries to send the most prioritized message. Received messages are inserted into the *msgReceived* table. As sending and receiving of messages is encapsulated in its own (Linux) process, it can be accomplished independently of other services. This guarantees the best usage of bandwidth and very good responsiveness.

#### 2.3.5. Service Query

The serviceQuery is a query response service. A user on ground can send a one-time query over the Iridium SBD to the database. The serviceQuery tries to answer it and generates a response message.

In detail, a user can send a one-time query request via email to the on-board device using the following format:
<id–service–query>:<priority>:<database> <SQL–statement>

Two example query requests are given below:
5:4:system.db SELECT ∗ FROM Log5:2:asd.db SELECT shipID FROM shape WHERE shipArea >= 50

The query request is received by the serviceSBD and saved in the *msgReceived* table. The serviceQuery checks the *msgReceived* table periodically. If query requests have arrived, the most prioritized is executed, and the query request is moved from the *msgReceived* table to the *msgReceivedArchive* table. The result of the one time is then put into an SBD message and inserted into the *msgToSend* table.

With serviceQuery, the user can access all on-board databases. As a typical example, the user can request a list of objects which have a defined size and have been detected within a defined time interval.

#### 2.3.6. Service Push

The servicePush is a messaging service that sends automatic notifications if a predefined event occurs. Events can be added and deleted during operation. Examples for such events could be the detection of oil near a ship (ocean dumping), ships entering a restricted area, ships sending no AIS signals, etc.

In detail, an event is defined as an SQL query with timing information. The timing information contains a time window and a period specifying the time points of execution of the SQL query. All activated events are saved in the *push* table. Events can be added or deleted by modifying the *push* table.

The following example shows how an event can be added to the *push* table via a query request:
5:3:system.db INSERT INTO PushTable(Start,Stop,Periode_s,Priority,Category,Db,Query) VALUES(‘‘2018-04-12 08:36:00’’,‘‘2018-04-12 20:45:00’’,‘‘300’’,‘‘5’’,‘‘107’’,‘‘asd_DB.db’’,‘‘SELECT shipID,course,speed FROM ships ORDER BY shipID DESC’’)

In normal words, within the time window, every 300 s, AMARO shall try to send information about the IDs, courses, and speeds of the latest detected ships. If the query is successful, servicePush generates a result message and inserts it into the *msgToSend* table.

#### 2.3.7. Service Ship Detection

The serviceShipDetect is responsible for data analysis. It receives the image data from the camera, analyzes them, and enters the results into a database table. Within the flight experiment, the image data are acquired with a frequency of 1 Hz (one acquisition per second) and are sent from the camera control computer to the AMARO system over ethernet. Since subsequent acquisitions will have overlapping content of around 90%, more than one observation will be made for one and the same object. For another mission with other conditions of image acquisition, these values may differ. The detected objects are examined and filtered out if they are too small or too big, or if one of the shape attributes does not match the defined constraints for being a ship. The considered shape attributes are: Size, perimeter, long axis, short axis, axes ratio, circularity, rectangularity, convexity, and solidity. More information about definitions and methods of calculation of these attributes can be found in [33].

If a ship-like object is detected in one acquisition, the following characteristics are extracted and stored in the database:Time stamp of each observation;location of each observation in geographic coordinates;shape attributes, as mentioned above.

Two ship-like objects are considered “similar” when they have both appeared within a limited geographical range and a limited time range, and when both have similar shape attributes, as defined above. If, in two or more subsequent acquisitions, “similar” ship-like objects are detected, they are grouped together and treated as a possible ship. The single objects are marked as assigned in order to not check them again. If, in at least four subsequent acquisitions, “similar” ship-like objects are detected, they are treated confidently as ships, and the following characteristics are extracted additionally:Number of observations,heading, andvelocity.

The object data can be directly accessed by the end-user via a query message (serviceQuery) or by defining an event (servicePush). In the current version, only the thermal channel was used. The computational steps involved are correction and normalization of the image data, water–land classification, connected component labelling [34], object analysis, and data comparison on the object’s metadata. For further reading, see [35]. As the data analysis is relatively complex and a high amount of data has to be processed, the serviceShipDetect can be run up to eight times in parallel.

### 2.4. MACS and Image Data

Images were acquired using the instrument MACS (Modular Aerial Camera System), cf. [36,37]. A picture of the MACS camera system can be found on Figure 5. Using the MACS camera, the photos were calibrated for radiometric correction and georeferenced, providing geographic coordinates, position accuracies, and absolute time for every image pixel. For the AMARO experiment, the system was equipped with a passive optical multi-sensor configuration to cover human-visible (RGB), near-infrared (NIR), and thermal infrared (TIR) spectra, as summarized in Table 1, but eventually, only the TIR channel was transmitted to the AMARO-Box. The image rate can be up to four full frames per second simultaneously for all sensors, and was set to 1 Hz during the flight experiment.

Through a hole in the aircraft fuselage, the lenses have an unobstructed view downwards. An embedded desktop class computer enables raw data recording, preprocessing, and immediate data forwarding. The MACS main computer is connected to the AMARO on-board computer through a Gigabit Ethernet link. Data of the selected image sensor are continuously fed as a byte stream. On this real-time stream, the object classification is executed in-memory, hence, without any image storage. Additionally, a function runs on the AMARO computer to re-establish geographic coordinates: Depending on the aircraft position and altitude the images are projected on sea level. The elevation of this plane is derived from the SRTM database. Because the scenery is completely flat over the sea, an image-edge four-point projection is sufficient. For a given image pixel, i.e., corresponding to a matched object, the function interpolates the edge coordinates and provides the geographic coordinates for the particular pixel.

## 3. Results

### 3.1. Experimental Flight

The experimental flight was conducted on the 12th of April in 2018. The AMARO-Box, the antennas for Iridium and the AIS, and the MACS camera were installed into a small science aircraft, a Cesna 207T, provided by the Freie Universität Berlin. The flight started from the airfield Schönhagen, located 50 km south of Berlin, Germany, at 09:15 a.m. UTC, and ended ibidem at 03:21 p.m. UTC. From there, the route lead over northern Germany to the mouth of the Elbe in Hamburg, where the actual experiment was conducted. The flight path is depicted in Figure 6.

In the time between 11:10 a.m. and 11:54 a.m. UTC, the main naval traffic route to enter the port of Hamburg was flown forward and backward (see Figure 7). This is called the experimental core time. Afterwards, the flight was interrupted to refuel the aircraft from 11:59 a.m. to 01:10 p.m. UTC. An overview of the different phases of the experimental flight is given in Table 2.

On-board were the pilot and two scientists, one to supervise the AMARO-Box, the other to control the MACS camera and to support the pilot. The supervision of the AMARO-Box was actually not necessary, since it was designed to operate autonomously. However, to be on the safe side for the first in-flight test, we considered supervision to be beneficial in case of unforeseen misbehavior. For controlling purposes, we connected the AMARO-Box with an external terminal PC. On-ground, two more people were assisting to install the camera and the AMARO-Box in the airplane.

The actual experiment—the communication with the AMARO-Box—was then conducted by a scientist and a technical assistant on-ground. Equipped with a standard office notebook, they operated the experiment from the user’s side in the airfield’s restaurant, which provided a stable internet connection. We want to mention that these users could have resided anywhere on Earth and could have used any device, as long as an internet connection was present.

### 3.2. Performance Communication

#### 3.2.1. Iridium Signal Quality

During operation time, the signal strength of the connection to the Iridium satellite network was measured and logged in the database *signal.db*. An evaluation of the database revealed an excellent overall reception quality for the whole flight. However, from about 10:30 p.m. to 11:00 p.m., no messages were received nor sent from the on-board AMARO-Box. In the evening, we received a notification from the Iridium SBD service informing us about unplanned intermittent outages which had taken place between 10:42 a.m. and 03:28 p.m. Since issues with the Iridium communication service also impact the AMARO performance in general, potential outages have to be taken into account when analyzing the performance. Nevertheless, it is worth noting that during the 15 months of using the Iridium SBD service, we received a total of five unplanned outage notifications, one of them just on the day of the experimental flight. The percentage distribution of the signal strength can be seen in Table 3. Figure 8a shows the distribution of the signal strength over time.

#### 3.2.2. Message Exchange

The first part of the operating time was taken by the flight to the experimental site at the North Sea. During that time, several messages were exchanged to establish and check the connection and to set up push queries.

In total, 56 messages were sent from ground to AMARO, while 169 messages were received from AMARO by the on-ground operator. From these, 13 and 34 messages are contemporary with the experimental core time, respectively.

Table 4 and Table 5 show the amount and type of messages sent from ground to AMARO and vice-versa. The push queries contained information about start time, expiry time, and period, i.e., the time interval in which the query should be executed by AMARO. One-time queries were executed as soon as possible after reception by the AMARO-Box on board. The possibility to exchange chat messages between the on-board and on-ground operators was set up in order to facilitate communication between on-board and on-ground operators during the flight. Empty downlink messages occurred due to technical reasons within the Iridium service, as described in ([38], Section 7.1.3).

Here, we give some examples for the message exchange during the experimental core time:Via a one-time query, AMARO was instructed to send the five latest log messages;via a push query, AMARO was instructed to send the number of hitherto acquired datatakes every 10 min;via a push query, AMARO was instructed to send the coordinates of the airplane’s current position every 12 min;via a push query, AMARO was instructed to send information about the latest 20 detected ships every five minutes;via one-time queries, AMARO was instructed to send information about objects with an area greater than 900 pixels and greater than 1800 pixels, respectively;via one-time queries, AMARO was instructed to send small quicklook images for several ships;via a one-time query, AMARO was instructed to send the MMSI for all ships inside a quadrilateral defined by latitudes and longitudes of its corners;via one-time queries, AMARO was instructed to send details for several MMSIs.

To all queries, a category is assigned. The answers are branded with the same category, such that the operator on-ground is able to match them with the corresponding queries.

#### 3.2.3. Query–Response Time Interval

Figure 8b shows the distribution of the time intervals between sending a query and receiving the corresponding answer during the experimental flight. It can be seen that these results coincide with the measurements of the SBD signal strength. Furthermore, Table 6 lists the number of message pairs (query/answer) with the time span between sending the query and receiving the answer.As the messages are sent and received by email, the time between the outgoing of the query and the incoming of the corresponding answer is measured with a temporal resolution of one minute. For time intervals larger than five minutes, the uplink time of the query and the downlink time of the corresponding answer were also analyzed. Note that the computation time is negligible, because the processing times of queries involved only database accesses.

For 7 out of the 46 query messages, we received no answer at all for different kinds of comprehensible reasons, e.g., due to an incorrect SQL syntax or a preceding delete-query where an answer is not expected. These messages are not taken into account hereafter. Apart from this, one message was answered with a delay of 68 min, where uplinking the query took 67 min and downlinking the answer took 1 min. As the query was sent during the refuel stop of the airplane, during which the AMARO system was deactivated, it is also not taken into account.

For 32 out of the remaining 38 messages, i.e., around 84%, the time span between query and answer was below five minutes, with an average of 1.87 min.

The response time for three messages (8%) was between 5 to 10 min, with an average of 6 min.

Another three messages (8%) were answered between 10 and 30 min. The average delay in this range was 20 min, with an average uplink delay of 18 min and an average downlink delay of 2 min. The most likely explanation for the high delays is the Iridium outage mentioned in Section 3.2.1, as the three queries in question were sent subsequently during the beginning of this time frame.

### 3.3. Performance AIS

During the operating time, 303,986 AIS messages were received by the system, with 275,144 AIS messages of types 1/2/3 and 7660 of type 5. A further 13,082 unsupported messages were received. A detailed overview is presented in Table 7.

In Figure 9, the aggregation over time of received AIS messages is displayed. AIS data were received during the complete operating time, except during the refuel stop, during which the AMARO-Box was not activated. All AIS messages were stored on-board in the AIS database, which was queried several times on the return flight. However, to match them with the results of the image processing is left for the next stage of expansion.

### 3.4. Performance Image Processing

As mentioned in Section 2.3.7, image processing was carried out on the thermal channel only. We abstained from creating a mature algorithm in terms of state-of-the-art remote sensing and Earth observation, since our main focus was to demonstrate a prototype for a globally deployable real-time information system. Nevertheless, the algorithm performed quite well. Apart from this, our service also includes the possibility of downlinking a quicklook of the object, such that an operator can double-check the result by visual inspection. An example set of quicklook images is displayed in Figure 10.

Due to the limited communication bandwidth, the maximal data volume of a quicklook was very limited. With a combination of a small image size, reduction of the color depth to one-bit monochrome, the use of a standard run length compression schema, and splitting up of the images into several parts, it was possible to fit the images in one to three SBD messages, each with a size of around 300 bytes.

During the whole experiment, 13,928 thermal images were acquired by the MACS sensor, while 13,607 images were processed by AMARO. Hence, 321 either got lost during transfer or were missed by AMARO because the processing channels were already busy. During the experimental core time, approximately 2570 thermal images were acquired, of which 25 were not processed. All results from the image processing thread were stored on-board in an SQlite database file. The results of the post-flight analysis are summarized in Table 8.

Actually, since the algorithm was designed for objects that are surrounded by water, the results during the flight over land are not meaningful. Therefore, the verification of the algorithm’s performance is done for the core time only. From the 26 results that were marked by AMARO as ships, we could verify by visual inspection that 23 were truly ships. Out of these, 13 were assigned one-to-one, i.e., AMARO detected one ship where we also see one ship in the images. An example of the visual inspection of one ship observation is shown in Figure 11.

It happened three times that AMARO detected two distinct ships in a time series of subsequent images, where only one and the same was present. In one case, AMARO detected two ships where there were two ships, but mixed up the results. Apart from this, AMARO re-detected three ships, i.e., these ships were overflown two times (while overflying the mouth of the Elbe forward and backward), and AMARO recognized them as one and the same object, which may be wanted or not, depending on the definition. If this effect is undesired, the time span for identifying “similar” objects could be narrowed further on. No ships were missed by AMARO compared to the visual inspection. For a quicker overview, these results are summarized in Table 9.

Even though the design of the algorithm was not our main focus, development efforts were kept comparatively low and only the thermal channel was used; the results are perfectly satisfactory. However, a thorough comparison with other ship detection algorithms would go beyond the scope of the present paper.

## 4. Discussion

We developed a comprehensive prototype system called AMARO for future real-time ship detection on-board satellites and other Earth observing vehicles. It includes on-board image processing, real-time communication via a satellite network, and a user-driven message exchange. To test the concept, the AMARO-box was built as prototype hardware, and the system was tested within a flight campaign over the North Sea.

### 4.1. Communication

Most special focus was put on the user-driven near-real-time information capability facilitated by using a satellite communication service. It was successfully demonstrated within our flight campaign, in which the Iridium SBD service was used for message exchange. More than 84% of the user queries were answered in less than five minutes with an average of less than two minutes.

For EO satellites, an information flow within minutes is not possible with the current approach of downloading the sensor data to ground stations and processing them on-ground. In contrast to conventional remote sensing missions, our system does not rely on any direct link to a ground station. By using satellite communication services, as demonstrated with AMARO, product information can be communicated to any device on the ground with connection to the internet, independent of the localities of both the carrier platform and the user. The system is therefore flexibly deployable at varying monitoring sites and especially suitable for the surveillance of remote areas without ground connection; for example, over the open sea. Especially for micro- and nano-platforms, this can be a feasible approach for enabling real-time capability, as it can be used worldwide, 24/7, and no ground infrastructure is required. Apart from this, the operational costs are affordable, even for smaller missions.

Apart from this, with AMARO, users are not drowned with an unmanageable amount data. They can control the flow of information by interactively interchanging messages with the on-board system. They can configure the automatic notification service during operation to get custom-tailored information about events of their interest. Finally, they can request further details by querying the on-board databases.

Being able to get information about ships within few minutes after observation, as we demonstrated, is beneficial in various situations. For example, it can support maritime safety agencies to take action against smuggling, illegal fishing, and sea pollution or support sea rescue services.

Nevertheless, regarding the communication procedure, some aspects were deemed to be in need of improvement. As described in Section 2.2.1, the queries in the SQL language were recorded in text documents and sent to AMARO as email attachments. The AMARO system replied the same way. It turned out that this procedure was uncomfortable to handle, even for the experienced operator. Requests and their corresponding answers always started with the same ID for an easier matching, but nevertheless, it was difficult to oversee which answers were already received, which were wrong, and which were empty or not present at all.

One of our priorities regarding further development is therefore the design of a graphical user interface. In principle, the interface should handle user-defined requests to a database via the internet. In the upcoming stage of expansion, every authorized user should be able to retrieve the information of their personal interest via a web application, using the device of their choice (smartphone, tablet, laptop, etc.). Apart from this, the limited bandwidth of around 300 B per message was a bottleneck in the communication flow. Sophisticated programming and workarounds were necessary in order to transmit a reasonable amount of information. The quicklook images could only be sent as highly compressed binary shapes. However, here we are sure that our approach will be augmented by ongoing and future development, which will continuously allow higher transmission rates. For example, with their next-generation satellites launched in the recent years, Iridium SBD can now transmit packages of around 2 kB in message size, compared to the previous 300 B. Furthermore, there may evolve even more possibilities with globe-spanning satellite-borne internet systems, such as OneWeb or StarLink.

Regarding the deployment on EO satellites, further investigation is necessary to examine the potential of the existing real-time communication services in LEO orbits. Satellite communication networks are usually designed for operating services on-ground and, hence, provide continuous coverage within their operational area on the Earth’s surface. Since EO satellites typically fly in an altitude of approximately 200 to 2000 km, at that height, coverage may be rather discontinuous. Apart from this, depending on the relative orbits of EO and communication satellites, a loss of connection may be encountered due to the amplified Doppler effect [39,40]. However, some on-board experiments were already conducted and yielded apparently promising results [41,42].

### 4.2. Onboard Data Analysis

Within AMARO, image data are processed directly on-board in order to extract the relevant and rather small-sized product data. In combination with using satellite communication, on-board data reduction is the prerequisite that enables real-time information.

Although the designed algorithm uses the thermal infrared channel only and is altogether kept relatively simple, the results were definitely competitive. More than 88% of the detected objects could be identified as ships. No ships, which were identified by eye, were missed.

We want to mention that this ship detection algorithm was primarily developed to be able to demonstrate the concept of a real-time onboard ship detection system in general. Only limited resources were available for the development and the validation of the ship detection algorithm. For a future version of the system however, we are planning to cooperate with remote sensing experts to integrate a mature, validated, state-of-the-art ship detection and classification processor.

Currently, we are part of the project ScOSA (Scalable On-Board Computing for Space Avionics), which has the goal of developing a high-performance on-board computer for satellite platforms [31]. The ScOSA system consists of multiple hardware nodes, uses a distributed computing approach, and can be dynamically reconfigured during runtime to remove faulty nodes and shift applications to healthy ones. We contribute to this project by porting AMARO to the ScOSA platform in order to stress the overall system and demonstrate its computing capacity [30].

It was not part of our experiment to synchronize the signals from the AIS receiver with the results from the image processing. However, the fusion of AIS and image data would bring a significant benefit. In particular, ships without signals could thus be identified. In the scientific community, there are several ongoing projects engaged in the fusion of AIS and image data [43]. Hence, we are establishing cooperations to rely on profound experience for the future improvement of our application. At this stage, we would like to mention that our system is not limited to optical data and AIS. Other sources of signals, e.g., an SAR camera (Synthetic Aperture Radar) or a pager for mobile phones, can be added without modifying the existing concept or the software structure.

### 4.3. System Design

Several publications about the individual subsystems exist, e.g., on-board image processing or real-time communication. However, our investigation and development is aimed at designing an operable system as a whole. We designed a comprehensive modular system for on-board data analysis and real-time information. It detects vessels and sends the results to the interested user within minutes after sighting. Our system is not designed as a monolithic block, but is flexibly expandable and deployable. It is modeled similarly to modern internet searching engines, consisting of a big database and several services that request and modify the database. The software system is therefore easy to expand, to adapt, and to maintain. AMARO is not set up as a simple one-way processing chain, i.e., getting images, extracting information, sending results. In fact, it is an autonomously working entity respondent to the user’s needs.

### 4.4. General Limitations of the System

By now, the main benefit of the system is achieved by using optical image data. Therefore, usability of the system heavily depends on the weather and lighting conditions. Operation at night is not supported, and during the day, heavy cloudiness can seriously limit the surveillance performance of the system. In the future, synthetic aperture radar sensors may be used to noticeably enhance the surveillance usability of the system. By now, this option is not feasible due to the weight and energy usage of available sensors and the high computing performance needed to process the data. Furthermore, with satellites, a permanent surveillance of a specified region is not feasible, as geostationary satellites do not provide a reasonable image resolution. However, in such a scenario, we see the benefit of the system as an additional data source, instead of as a single permanent surveillance solution.

### 4.5. Expansion of Deployment

A field to be investigated in more detail is that of possible flight devices. High-altitude pseudo-satellites seem to be predestined for this, since they offer the possibility of continuously monitoring an area of interest autonomously and for a longer duration. With the DLR working on the development of a high-altitude platform [44] and commercial systems like the Airbus Zephyr [18] starting to become available, we think that, within the next five years, suitable flight platforms may be a realistic option.

Finally, we are planning to expand our system to be deployable for other time-critical Earth observing scenarios that would benefit from a rapid information system; for example, real-time monitoring of traffic, sea ice, or disasters.

## Figures and Tables

**Figure 1 sensors-20-01324-f001:**
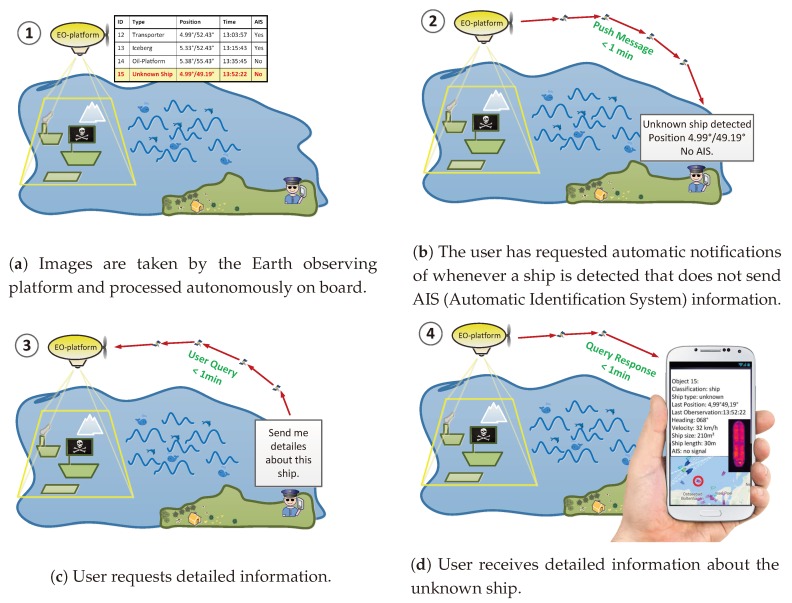
An exemplary user story.

**Figure 2 sensors-20-01324-f002:**
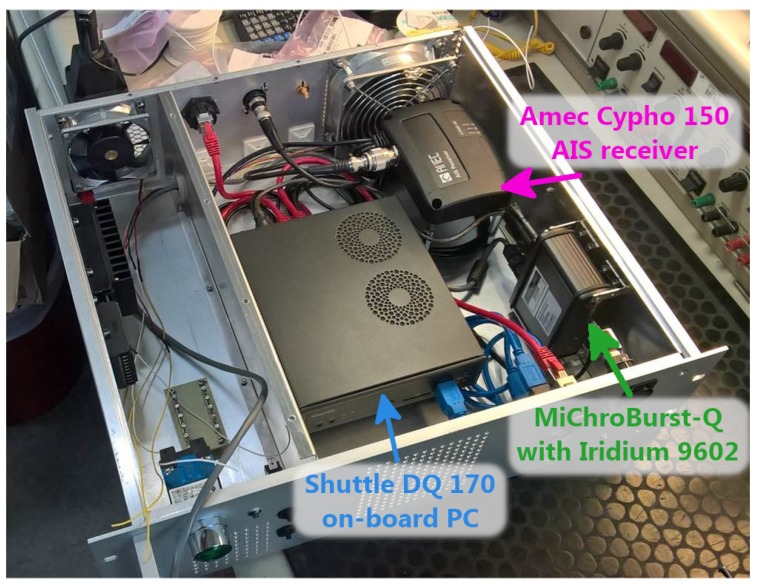
Autonomous Real-Time Detection of Moving Maritime Objects (AMARO)-Box with hardware components during assembly.

**Figure 3 sensors-20-01324-f003:**
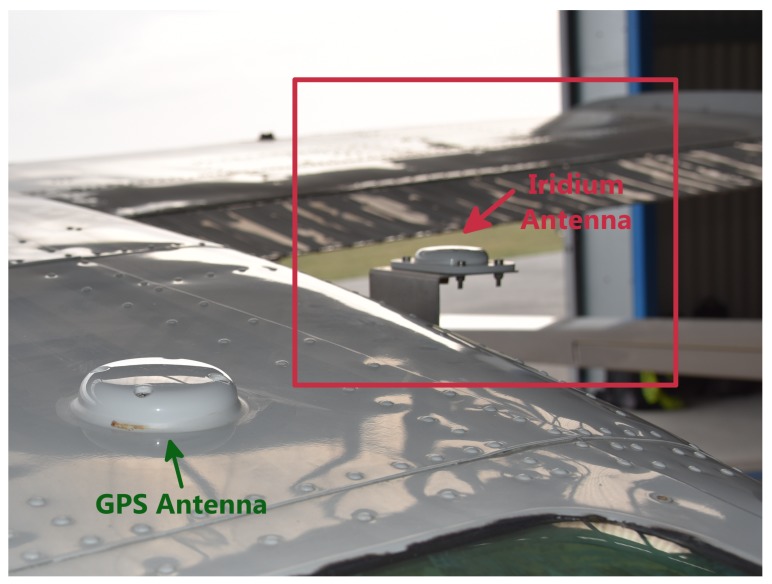
AeroAntenna AT2775-110 Iridium Antenna mounted on the airplane’s roof.

**Figure 4 sensors-20-01324-f004:**
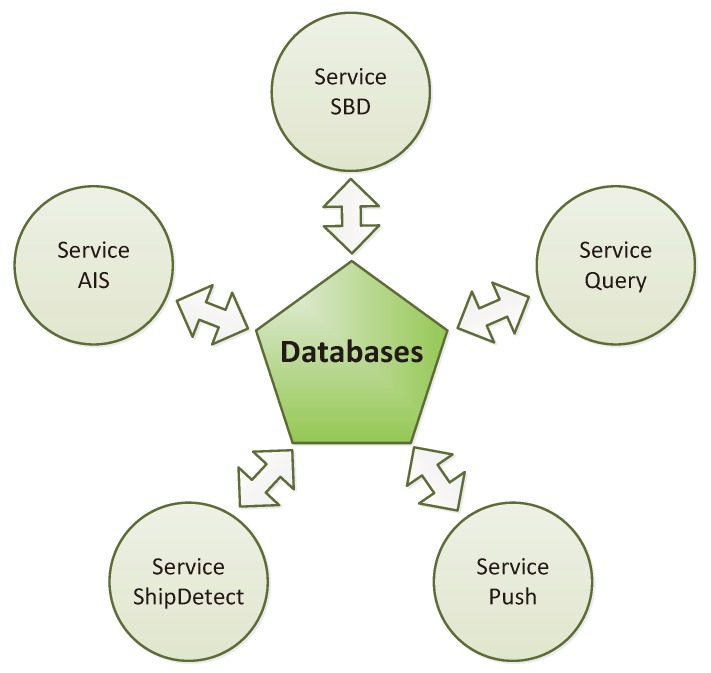
Top level overview of the AMARO software architecture.

**Figure 5 sensors-20-01324-f005:**
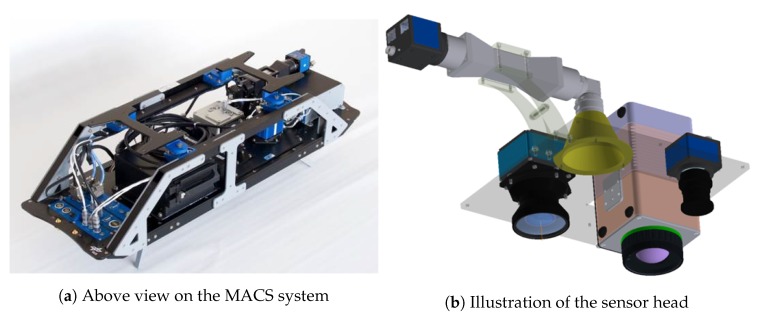
Modular Aerial Camera System (MACS).

**Figure 6 sensors-20-01324-f006:**
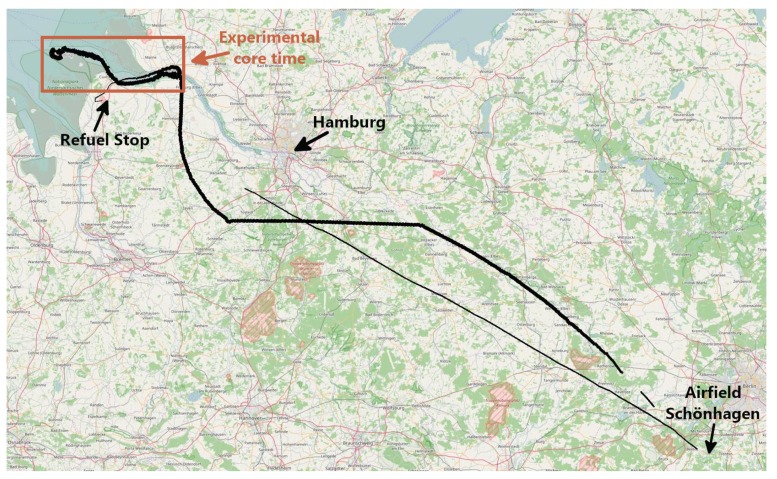
Flight path from the airfield Schönhagen to the North Sea and back.

**Figure 7 sensors-20-01324-f007:**
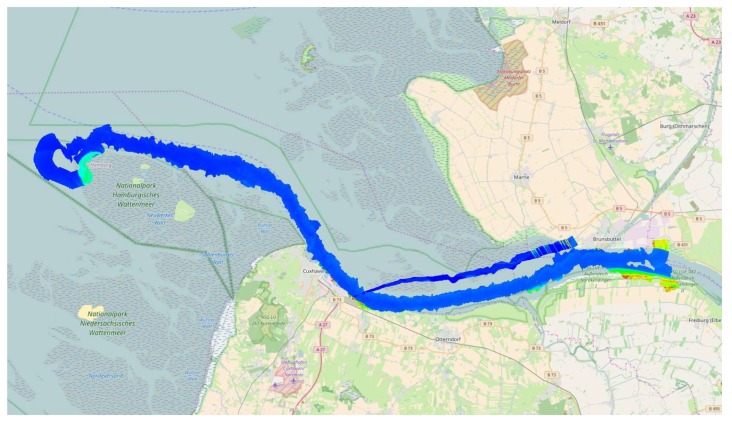
Mosaic of thermal images over the mouth of the Elbe during AMARO’s experimental core time.

**Figure 8 sensors-20-01324-f008:**
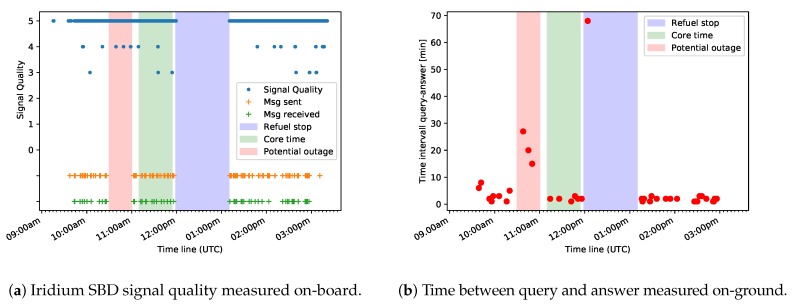
Iridium Short Burst Data (SBD) signal and response measurements.

**Figure 9 sensors-20-01324-f009:**
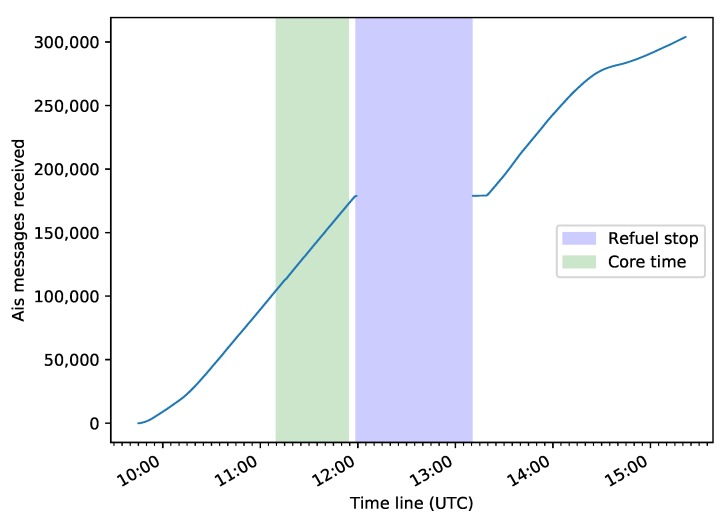
Aggregation of received AIS messages during the operating time.

**Figure 10 sensors-20-01324-f010:**
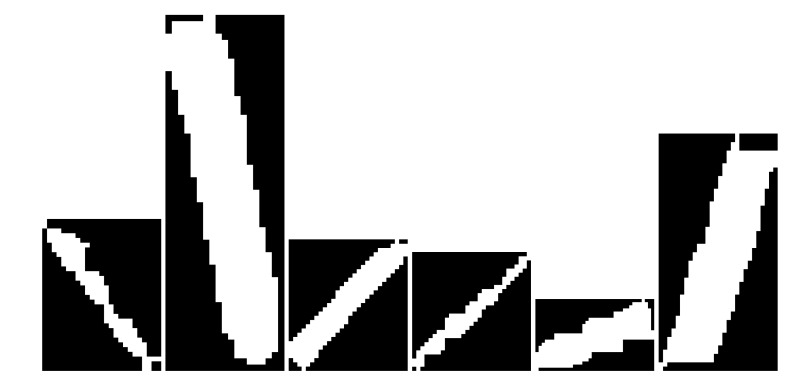
Example of quicklook images of potential ship objects.

**Figure 11 sensors-20-01324-f011:**
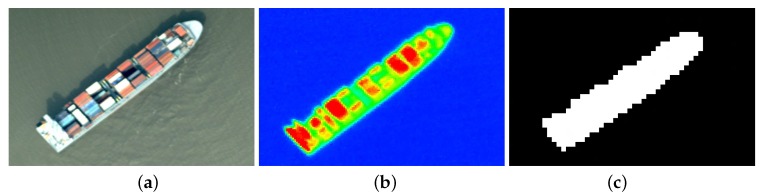
Ship detected by AMARO, 12th April 2018, 11:45 am UTC, Mouth of the Elbe, Hamburg, Germany: (**a**) RGB image (**b**) thermal image (**c**) quicklook which was sent to ground from AMARO.

**Table 1 sensors-20-01324-t001:** Modular Aerial Camera System (MACS) sensor setup.

	RGB (Bayer Color Pattern)	Near Infrared	Thermal Infrared
Spectral Bands (nm)	400–520 (blue) 500–590 (green) 590–680 (red)	700–950	7500–14,000
Resolution (pixels)	4864 × 3232	3296 × 2472	1024 × 768
Focal length (mm)	50.0	29.3	30.0
Pixel pitch (μm)	7.4	5.5	17.0
GSD @ 820 m above sea level (cm)	12.1	15.4	46.5
GSD @ 2500 m above sea level (cm)	37.0	49	141.7
Field of view across track (deg)	39.6	34.6	32.4

**Table 2 sensors-20-01324-t002:** Overview of the timing of the different phases of the experimental flight.

Time Intervals	UTC	Local Time (MESZ = UTC + 2 h)
Flight time	09:15 a.m.–03:21 p.m.	11:15 a.m.–05:21 p.m.
Refuel stop	11:59 a.m.–01:10 p.m.	01:59 p.m.–03:10 p.m.
Operating time	09:15 a.m.–11:59 a.m. 01:10 p.m.–03:21 p.m.	11:15 a.m.–01:59 p.m. 03:10 p.m.–05:21 p.m.
Experimental core time	11:10 a.m.–11:54 a.m.	01:10 p.m.–01:54 p.m.

**Table 3 sensors-20-01324-t003:** Distribution of Iridium’s signal strength over time in [%] measured on-board.

		Distribution During	Distribution During
Signal Strength	Signal Strength	Operation Time [%]	Core Time [%]
0	no signal	0%	0%
1	very low	0%	0%
2	low	0%	0%
3	medium	0.63%	0.35%
4	strong	1.65%	0.70%
5	very strong	97.72%	98.94%

**Table 4 sensors-20-01324-t004:** Number of uplinked messages.

Message Type	Operating Time	Core Time
One-time queries	43	9
Push queries	6	2
Delete instructions	5	1
Chat messages	2	0
Total	56	13

**Table 5 sensors-20-01324-t005:** Number of downlinked messages. In M1 messages, the answer fits into one single packet, while for >M1, it had to be split into several parts.

Message Type	Operating Time	Core Time
Answers to one-time queries M1	26	3
Answers to one-time queries >M1	23	3
Answers to push queries	55	16
Empty messages	50	8
Chat messages	15	4
Total	169	34

**Table 6 sensors-20-01324-t006:** Query–Response time: Time interval between sending a query and receiving the corresponding answer (email to email). Gray values were not taken into account for further analysis.

Time Span [min]	Average Uplink/Downlink Time [min]	Contacts
1	-	10
2	-	16
3	-	6
5	1/4	1
6	4/2	1
8	2/6	1
15	13/2	1
20	18/2	1
27	25/2	1
68	67/1	1 (refuel stop)
-	-	7 (no answer expectable)

**Table 7 sensors-20-01324-t007:** Overview of AIS messages received on-board. For a detailed description, see [26].

Type	Count	Percentage	Description
raw	303,986		AIS data frames
all	295,886	100%	Supported and unsupported messages
supported	282,804	96%	
123	275,144	93%	Position report
5	7660	3%	Static and voyage related data
not supported	13,082	4%	
4	3737	1%	Base station support
6	1	<1%	Binary addressed message
7	9	<1%	Binary acknowledgement
8	5161	2%	Binary broadcast message
9	9	<1%	Standard SAR aircraft position report
10	2	<1%	UTC/date inquiry
11	135	<1%	UTC/date response
15	971	<1%	Interrogation
17	31	<1%	DGNSS broadcast binary message
18	1223	<1%	Standard Class B equipment position report
20	129	<1%	Data link management message
21	886	<1%	Aids-to-navigation report
23	42	<1%	Group assignment command
24	732	<1%	Static data report
27	14	<1%	Position report for long range applications

**Table 8 sensors-20-01324-t008:** Analysis of the results produced by the ship detection thread.

		Full Flight Time	Experimental Core Time
Objects	total	144,988	12,860
	ship-like	2339	324
	assigned to ship	721	294
Ships	total	188	47
	category active ship	68	26
	category initialized ship	120	12

**Table 9 sensors-20-01324-t009:** Comparison of the results from AMARO with visual inspection.

Description	Number
detected by AMARO	26
unambiguously identifiable by human eye	23
results from AMARO and visual inspection assignable one-to-one	13
AMARO detected two ships where only one was present	3
AMARO mixed two distinct ships	1
AMARO re-detected ships on the return flight	3

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
