# Peer review of "AMARO—An On-Board Ship Detection and Real-Time Information System"

_sensors, 2020, doi:10.3390/s20051324_

Round 1
Reviewer 1 Report
This manuscript is written and organized well. Only one minor issue should be noticed before acceptance. As a ship detection method, please provide more detection results for the readers.
Reviewer 2 Report
L120 which he gets...
L173 off the shelf
L186-L189 this informatiin is easily outdated and less relevant
L198 define class A and class B messages (or provide a reference)
L275 check wording "severally"
L374-376 pleae clarify if this is an inherent part of the MACS system
Table 1: GSD should be in the list of acronyms?
Sec 3.2.1 Comment: it appears that Iridium outage can be a potential limitation to the effectiveness of your system. PLease consider to include this also in the discussion in chapter 4.
L505: please expand on this statement.
L569: colloquialism "pretty uncomfortable", should be avoided
L633-635 PLease expand or provide a reference to your plans. (As such, the statement is irrelevant to the paper.)
Reviewer 3 Report
This is a feasibility study of an autonomous ship detection system and as such, quite well-written and results are sufficient. I think this paper will be of high interest to a few readers/organizations who are invested in ship surveillance, hence I put the overall merit to high.
The only places I would recommend minor edits are:
Introduction:If possible, will be good to have better review of similar feasibility studies, though I understand that is not always published research. After all, autonomous ship surveillance is hardly a new concept, sonar surveillance systems do it all the time. Any developed country with naval dominance has some form of autonomous surveillance of shipping traffic, even if such surveillance is more covert.
In the discussion of results it will be good to include where the feasibility study will not apply, e.g. when will the proposed hardware/software systems be inadequate? The paper discussed in depth AIS and EO-based surveillance. It will also be important to state the degree to which the proposed system is designed to interface/complement remote surveillance systems, e.g. when a detection is made that is not verified via satellite imagery, what is the (potential) false alarm resolution in this case? Similarly, to what degree is the type of vessel detected verifiable via third-party human confirmation? The authors allude to deployment of of their method in EO satellites (lines 535-539) but as far as I could see, no effort is proposed in reconfirmation via independent sources to keep the false alarm rates low. Maybe they can comment on how they expect to keep false detection (both vessel detection rate and type of vessel detection rate) in check?
